# Auricular Non-Epithelial Tumors with Solar Elastosis in Cats: A Possible UV-Induced Pathogenesis

**DOI:** 10.3390/vetsci9020034

**Published:** 2022-01-18

**Authors:** Francesca Millanta, Francesca Parisi, Alessandro Poli, Virginia Sorelli, Francesca Abramo

**Affiliations:** Department of Veterinary Sciences, University of Pisa, Viale delle Piagge n. 2, 56124 Pisa, Italy; francesca.millanta@unipi.it (F.M.); alessandro.poli@unipi.it (A.P.); virginiasorelli@yahoo.it (V.S.); francesca.abramo@unipi.it (F.A.)

**Keywords:** auricular, cats, elastosis, neoplasms, solar, UV light

## Abstract

The photoinduced etiopathology of actinic keratosis and squamous cell carcinoma in feline species is well known. This etiology has also been reported for non-epithelial cutaneous tumors in other species. To date, no cases of auricular non-epithelial cutaneous neoplasms erased in a contest of actinic keratosis in cats have been reported. The aim of this study was to describe feline auricular non-epithelial cutaneous neoplasms associated with typical UV-induced cutaneous lesions and solar elastosis. The study was conducted on five feline cases diagnosed with auricular non-epithelial cutaneous tumors (two fibrosarcomas, one mixosarcoma, one epithelioid melanoma and one hemangiosarcoma), selected from the Tumor Registry of the Department of Veterinary Sciences of the University of Pisa (1998–2018). Ten and six feline auricular biopsies of normal skin and skin with actinic keratosis, respectively, were used as controls. Orcein stain was used to investigate solar elastosis. Histological changes related to chronic solar irradiation were documented in the skin adjacent to the neoplastic lesions in the five cats. Considering the anatomical localization and the results of histopathology, this study suggests that non-epithelial cutaneous neoplasms may have a UV-induced etiopathogenesis in the feline species.

## 1. Introduction

The cutaneous barrier is continuously challenged by environmental hazards, the most ubiquitous of which is sunlight. It is generally known that chronic exposure to ultraviolet (UV) light is harmful and causes photodamage in humans and other animals [1,2,3]. The recent global climate changes result in increasing UV radiation [4], and the predictable consequence of this trend is an increase of solar-induced pathologies, making them an emerging topic of common interest. In humans, the effects of long-term UV exposure are known as photoaging [5]; UV-induced lesions occur mainly on sun-exposed body sites, such as the face, head, neck or extremities, and Caucasians with fair skin, and light eye and hair color are at increased risk [6,7]. Fur acts as a physical barrier to UV light, so UV-associated lesions arise more frequently on sparsely-haired, non-pigmented areas [8,9,10]. The common clinical changes associated with photoaging include dyspigmentation, laxity, yellow hue, wrinkles, telangiectasia, a leathery appearance and a variety of benign, premalignant and malignant neoplasms [11]. Histologically, photoaged skin is characterized by alteration in thickness, with areas of severe atrophy and/or hyperplasia, in pigmentation and in the degree of nuclear atypia of both keratinocytes and melanocytes. Moreover, photoaged skin shows loss of epidermal polarity and orderly maturation of keratinocytes that, when damage is severe and the cellular reparative mechanisms fail, undergo apoptosis with the formation of the so-called sunburn cells, considered a histologic marker of long-term UV exposure. Heliodermatitis is the typical change occurring in the dermis, characterized by numerous and hyperplastic fibroblasts, abundance of inflammatory cells, decreased type I and III collagen and increased elastin, in proportion to the amount of sun exposure. The photoinduced elastin appears to be abnormal, probably because of the effects of chronic inflammation, and occupies the areas previously held by collagen. The accumulation of thickened, tangled, and ultimately granular amorphous elastin fibers in the extracellular matrix are known as elastosis. Blood vessels in the elastotic dermis are often tortuous and dilated, with marked thickening caused by concentric deposition of basement membrane-like material [5,11,12].

A cause-effect relationship with exposure to UV light has been clearly proven for squamous cell carcinoma and its preneoplastic precursor lesion, the actinic keratosis [1,9,13,14]. Recently, some case reports suggested a contribution of UV light in the development of human cutaneous dermal sarcomas [6,7,15,16], vascular tumors [17] and melanomas [18,19]; of canine cutaneous [20,21,22] or conjunctival hemangiomas and hemangiosarcomas [23]; of equine ocular, periocular and vulvar hemangiosarcomas [24,25]. In cats, the onset of hemangiosarcomas on areas of the body that are mainly exposed to solar radiation and in fairly pigmented skin have also suggested a role of UV rays in the pathogenesis of tumors in this species [26,27]. Moreover, the domestic cat is reported to be a good model to study photodamage because spontaneous feline solar keratosis has a high incidence, and cats usually share the environment with their owners. In this perspective, cats can act as sentinels for human cancer development [8,9,28]. The aim of the present study was to document feline non-epithelial cutaneous tumors of the auricular pinnae, one of the sites most associated with UV-induced lesions, to investigate a hypothetical causative role of UV light in the development of these neoplasms.

## 2. Materials and Methods

### 2.1. Case Selection

A total of 11 biopsies of feline auricle samples were selected from the Tumor Registry of the Department of Veterinary Sciences of the University of Pisa (1998–2018). In detail, five samples from cats with a diagnosis of non-epithelial cutaneous tumor and six from subjects with actinic keratosis (control group 1) were included. Moreover, from ten dermatologically healthy stray cats that were referred for necropsy at the local veterinary hospital, normal skin samples were obtained from the auricular pinna (control group 2). This research was carried out according to the International Regulation governing the use of animals for scientific purposes (Directive 2010/63/EU). Institutional Ethical Committee Approval was not required because skin samples were obtained from necropsies or referred for diagnostic purposes.

### 2.2. Clinical Data

Clinical data regarding sex, age, breed, coat color and clinical abnormalities before and at the date of initial presentation of cases with pathological alteration of skin were collected.

### 2.3. Histopathology

Tissue samples were fixed for at least 24 h in 10% neutral buffered formalin, dehydrated through graded alcohols, clarified in xylene, embedded in paraffin wax, sectioned 5 μm thick, mounted on glass slides and stained with hematoxylin and eosin (HE). On HE-stained sections from each case, the following histological changes characteristic of chronic solar irradiation were recorded [1,9,29]: (a) epidermal hyperplasia, (b) epithelial stratification disorders, (c) presence of sunburn cells, (d) squamatization of the basal layer, (e) presence of proliferation nests, (f) presence of keratinocyte dysplasia, (g) abnormal parakeratosis, (h) presence of elastosis, (i) presence of mitosis. To better investigate solar elastosis, Orcein staining was performed following the manufacture’s instruction (Orcein kit, Bio-Optica, Milano, Italy). The elastic fiber abnormalities were evaluated through qualitative analysis.

## 3. Results

### 3.1. Signalment, Clinical Signs, Tumor Location

All five cats with non-epithelial cutaneous tumors were European shorthair, 3/5 had white coat and 2/5 had pigmented coat; 3/5 were neutered male, 2/5 were female; median age was 8.4 years (ranging from 5 to 11). The most common clinical signs reported at the time of presentation were the presence of a nodular lesion (4/5) or crusts (1/5) on the auricular pinna.

The subjects of control group 1 were all European shorthair with white coat; 3/5 female and 2/5 male cats; median age 8.25 years (ranging from 6 to 10). The most common clinical signs at the time of presentation were erythema, scaling and crusts. Control group 2 included all European shorthair cats, 5/10 with white coat, the other 5/10 with pigmented coat. No other data about control group 2 were available since the samples belonged to stray cats.

### 3.2. Histologic Evaluation

The group of cats with non-epithelial cutaneous tumors of the auricular pinna included two subjects with fibrosarcomas, one subject with a mixosarcoma, one case with an epithelioid melanoma and one with a well-differentiated cutaneous hemangiosarcoma.

Both fibrosarcomas were nodular lesions characterized by the proliferation of interlaced bundles of fibroblasts with interposition of collagenous fibers. The neoplastic cells were characterized by elongated, oval or spindle-shaped nuclei, scant cytoplasm and one or more prominent nucleoli. UV-induced alterations, such as hyperplasia, squamatization, stratification disorders and proliferation nests were observed in the epidermis, while very little elastotic material accumulated in the dermis at the peripheral area of the tumors (Figure 1A).

The myxosarcoma was a dermal nodular lesion characterized by pleomorphic, starry cells scattered randomly within a finely granular basophilic amorphous myxomatous matrix. The abnormal fibroblasts had scant eosinophilic cytoplasm and pleomorphic nuclei, often hyperchromatic. Epidermal UV-induced damages, like hyperplasia, stratification disorders and proliferation nests were detected at the periphery of the lesion.

The epithelioid melanoma was a nodular lesion consisting of the proliferation of neoplastic melanocytes replacing large portions of the collagen in the dermis. These epithelioid cells had large nuclei, prominent nucleoli, a moderate to extensive amount of cytoplasm and fairly clear cell boundaries. The amount of pigment within the cytoplasm was mild to abundant. Elastosis was evident in the dermis; no other UV-induced perilesional changes were observed in the epidermis (Figure 1C).

The well-differentiated cutaneous hemangiosarcoma was characterized by endothelial cells delimiting an anastomotic network of blood-filled channels. Neoplastic endotheliocytes showed poor cytoplasm, large hyperchromatic nuclei, and scarcely prominent nucleoli. Alterations due to photoinduced damage, namely hyperplasia, stratification disorders and keratinocytes dysplasia, were found in the adjacent areas of the skin.

In the group of cats with actinic keratosis (control group 1), the epidermis showed mild to marked irregular hyperplasia. Two cases showed laminar or compact hyperkeratosis and parakeratosis. Mild to moderate stratification disorders were found in 5/6 cases, mainly represented by loss of polarity of the keratinocytes in the basal and spinous layers. Atypia of keratinocytes was characterized by cellular enlargement, hyperchromasia, nuclear pleomorphism and nucleolar prominence. Atypical keratinocytes showed a slightly lighter cytoplasm than the normal ones in the adjacent epidermis. Mitoses were found above the basal cell layer. Stratification disorders, such as nests of proliferation in the basal layer, were described in two cases. The presence of eosinophilic apoptotic sunburn cells was highlighted in four cases. The basement membrane was intact in all samples. There was dermal fibrosis in 2/6 cases (Figure 1E).

In the group of cats with a healthy auricle (control group 2), the pinna was covered on both sides by normal skin containing hair follicles and glands. Melanin pigment was described in the epidermis and hair follicles of the five subjects with black or brindle coats, while it was absent or rare and scattered in subjects with white coats. The basement membrane marked the transition from the epidermis to the dermis, which was made up of fibroblasts, few perivascular and interstitial mast cells and an extracellular matrix. The papillary dermis, difficult to identify due to the absence of the rete ridge in feline species, consisted of thin bundles of collagen fibers and abundant fundamental substances and capillaries; below this, the reticular dermis was identified by the presence of collagen fiber arranged in large bundles parallel to the skin surface (Figure 1G).

By Orcein stain, elastosis was detected in all samples of cats with non-epithelial cutaneous tumors. In all cases, the elastic fibers in the superficial dermis were thickened, fragmented, wavy to branched and tangled, losing the regular structure and orientation typical of normal skin fibers and sometimes aggregating into clusters (Figure 1B). The epithelioid melanoma showed a few areas of the dermis containing thickened, fragmented, knotted elastic fibers (Figure 1D).

The evaluation of special staining in control group 1 highlighted the presence of elastosis in 4/6 cases. In these cases, the elastic fibers appeared morphologically short, fragmented, knotted, with a non-linear, curved or sometimes jagged pattern (Figure 1F).

The evaluation of Orcein staining of the samples from control group 2 highlighted the presence of uniform, straight and linear filamentous elastic fibers, arranged parallel or oblique to the skin surface. Thinner and branched but shorter fibers were present in the superficial dermis, while longer fibers were observed in the interfollicular dermis (Figure 1H). There was no significant difference between the cats with white coats and the subjects with black or brindle coats, neither in the quantity nor in the morphology of the elastic dermal fibers.

## 4. Discussion

Tissue samples collected for this study came from auricular pinnae, an area directly exposed to UV light. These areas, together with the nose, eyelids and temporal regions, have been frequently associated with UV-induced cutaneous preneoplastic and neoplastic lesions, such as solar keratosis and squamous cell carcinoma [9,10]. Although the sample size of our study is not enough to draw conclusions, the trend reflects what has already been seen in humans and dogs, in which UV-induced lesions arise mainly on sparsely pigmented subjects, testifying that melanin pigment has a protective role in absorbing and scattering detrimental UV rays, as already stated in the literature [11]. The same considerations should be made regarding the age and sex of cats with non-epithelial cutaneous tumors. In human and canine species, tumors of similar origins were mainly related to adult subjects without predilection of sex [14,20,23].

Histologically, cases with non-epithelial cutaneous tumors showed many of the lesions classically described in photodamaged skin in tissues adjacent to the neoplasms. These were either found within the epidermal layer (hyperkeratosis and/or parakeratosis, mild to severe hyperplasia, stratification disorders, squamatization of epidermal basal cells, atypia or apoptosis of keratinocytes) and in the dermis (fibrosis and elastosis). Similar histological features are described in the photodamaged skin of humans and are related to the type and amount of UV radiation [5,11,30]. Interestingly, solar-induced lesions observed in the skin adjacent to the epithelioid melanoma were limited to the dermis (solar dermatosis) without the typical epidermal involvement seen for actinic keratosis [9]. This could be linked to the total amount of UV radiation that reaches the skin, which is directly influenced by the quantity of hair, amount of cutaneous pigment, and duration of exposure [9]. Moreover, a predominance of a particular wavelength of radiation may be implicated for the dermal, rather than epidermal, solar damage. Terrestrial solar radiation is a combination of UVB (290–320 nm) and UVA radiation (320–400 nm) [3]. UVB rays are highly energetic but reach the earth in low amounts, while UVA radiation is the most prevalent component of solar UV radiation (90% of UV rays reaching the ground), is lower than UVB in intensity, but it is much more abundant; therefore, UVA radiation plays a major role in causing solar damage. UVB radiation is almost entirely absorbed by the epidermis (70% from the stratum corneum), and only 10% penetrates the dermis. On the contrary, UVA radiation reaches the epidermis, where it is only partly absorbed, and 20–30% of it penetrates deeper than UVB radiation. Due to the differences in the components of absorption, UVB radiation has major effects on the epidermis if compared to UVA radiation [2]. For this reason, UVB rays mainly induce squamous cell carcinomas and actinic keratoses as a precursor, while UVA radiation can contribute to the carcinogenicity of sunlight, inducing profound alterations of the dermal connective tissue [2,31]. In this context, it appears reasonable that the lesions in the tissue adjacent to epithelioid melanoma could be relegated exclusively to the dermis, probably because the skin of that subject was reached mainly by UVA rather than UVB. Another possible pathogenic hypothesis could be that UVA radiation comes through glass, while, on the contrary, UVB rays do not. Thus, high UVA amounts can be received by indoor cats [2]. Even if solar elastosis has been considered as a rare manifestation of UV damage in feline species, due to the reduced amount of elastic fibers compared with humans [6], in our study, every subject with non-epithelial cutaneous tumors showed elastosis in the dermis adjacent to the neoplasm, confirming the previous findings of Mills et al. [26], regarding feline dermal hemangiosarcomas. Experimental studies on mice demonstrated that UV radiation produced severe elastic fiber hyperplasia and then degeneration of the elastin matrix, with an accumulation of thickened, degraded fibers and elastosis, as already described in humans [32,33,34]. Of note, in this study, elastosis was detected; however, this lesion can be subtle and easily overlooked during a routine examination. The Orcein stain my help in demonstrating this solar-induced alteration. Solar elastosis is considered a hallmark of photodamage [8,32,33,34,35] and has rarely been described in unexposed skin [5,8]; therefore, the involvement of UV rays in the carcinogenetic process of these tumors may be suspected. This hypothesis had already been suggested by other authors who described the association between elastosis and non-epithelial cutaneous tumors in other species: vulvar epithelial and ocular hemangiosarcoma in horses [24,25], canine cutaneous [20,21,22] or conjunctival hemangiomas and hemangiosarcomas [23], human cutaneous sarcoma [6,7,15,16], vascular neoplasms [17] and melanomas [18,19]. Moreover, previous studies about dermal hemangiosarcoma in feline species reported the ear tip as a typical location of these tumors only when UV radiation was suspected to be a causative factor in these neoplasms [26,35]. After all, the International Agency for Research on Cancer (IARC) [36] concluded already in 1992 that there was “sufficient evidence in humans for the carcinogenicity of solar radiation”, classifying ultraviolet radiation (UVR) as a Group 1 carcinogen, and that “Solar radiation causes cutaneous malignant melanoma and nonmelanocytic skin cancer”. Nowadays, it is accepted that overexposure to solar UVR is the main underlying cause of skin damage that, if sustained, can lead to skin cancer [5].

Besides the putative role of UV radiation in the carcinogenetic process of non-epithelial cutaneous tumors, Berwick et al. suggested that the association of these neoplasms with UV-induced lesions may also be relevant for prognosis [37]. Indeed, they observed that human melanoma might be biologically benign if in association with sun exposure, and in the veterinary field, solar-induced dermal hemangiosarcoma in dogs was suspected to be less aggressive than dermal hemangiosarcomas of other etiology [22].

The cause of these differences is not yet fully clear, but some hypotheses have been made. A protective role of UV rays was suggested through their involvement in the synthesis of 25-hydroxy vitamin D3 in the skin, which, when converted to 1,25 (OH)2D with antiproliferative and proapoptotic effects. Another possible explanation suggested was linked to the induction of melanization and the increase in DNA repair capacity typical of sun exposure [37]. In this perspective, the evaluation of solar elastosis has acquired increasing importance as a prognostic factor [38], and it would be useful to associate special staining highlighting elastin fibers, like the Orcein staining used in this study, in order to not underestimate elastosis.

Finally, domestic animals have been considered for a time as sentinels for human environmental carcinogenic agents [8,39] and models of human neoplasms [40]. Particularly, companion animals have a valuable role as sentinels if compared with the other species serving as animals sentinels because they share similar household and recreational risk factors as the people who own them [28,41]. Since cats closely share the environment with humans, studies on the main photodamage-associated feline neoplasms could be relevant not only in the veterinary field but also in comparative medicine.

## 5. Conclusions

Non-epithelial cutaneous neoplasms in our cats shared sites of predilection with solar-associated tumors and, like them, occurred in white-haired skin. The presence of epidermal and dermal solar-induced damage in the skin adjacent to the non-epithelial cutaneous tumors described in this study, located in areas usually involved by other classical epithelial solar-associated tumors and in white-haired cats’ skin, substantiate a prolonged exposition to UV radiation. Further studies should be encouraged to understand the possible role of UV light on non-epithelial cutaneous neoplasms and to study their possible prognostic significance.

## Figures and Tables

**Figure 1 vetsci-09-00034-f001:**
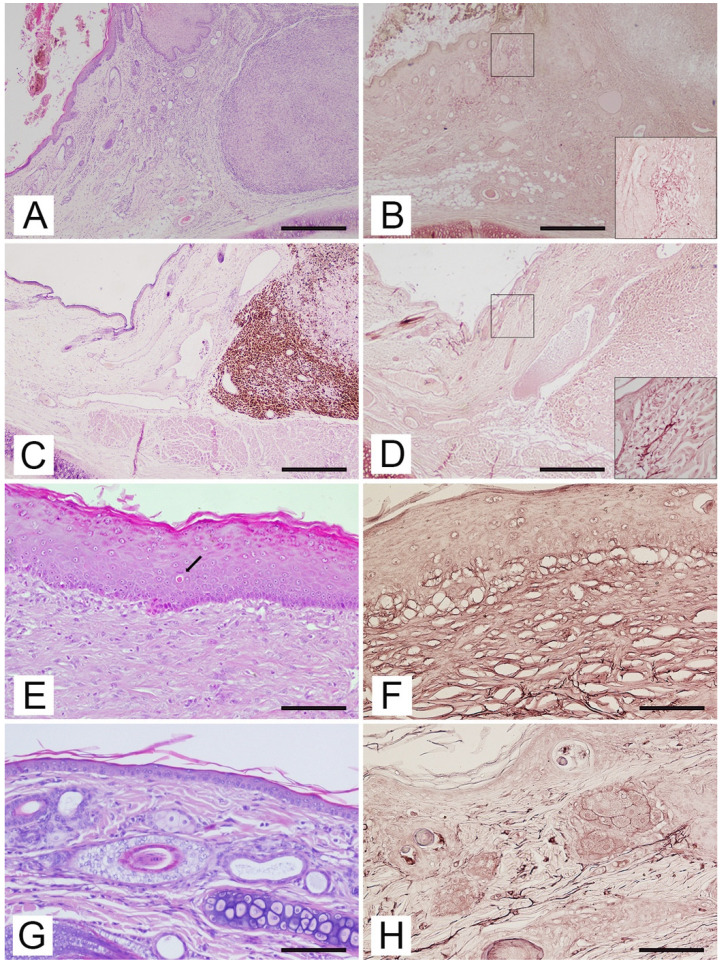
Auricular pinna, cat. (**A**) Fibrosarcoma. Hyperplastic epidermis with stratification disorder and dermal fibrosis in tissues adjacent to the neoplasm. HE. Bar = 50 µm. (**B**) Fibrosarcoma. Superficial dermis adjacent to the neoplasm with a considerable amount of elastic fibers. Insert: Higher magnification of hypertrophic, fragmented, branched or elastic fibers arranged in a dense cluster around a glandular structure. Orcein staining. Bar = 50 µm. (**C**) Epithelioid melanoma. Normal configuration of the epidermis at the periphery of the tumor. HE. Bar = 50 µm. (**D**) Epithelioid melanoma. Superficial dermis adjacent to the neoplasm with a moderate amount of hypertrophic elastic fibers. Insert: Higher magnification of hypertrophic, fragmented and tangled elastic fibers. Orcein staining. Bar = 50 µm. (**E**) Control group 1. Epidermal hyperplasia with increasing dermal collagen in horizontal arrangement and hypertrophic elastic fibers. Presence of a sunburn cell with pyknotic nuclei and a strong eosinophilic cytoplasm (arrow). HE. Bar = 200 µm. (**F**) Control group 1. Superficial dermis containing hypertrophic, branched, tangled elastic fibers. Orcein staining. Bar = 200 µm. (**G**) Control group 2. Normal configuration of the epidermis and dermis. HE. Bar = 100 µm. (**H**) Control group 2. Superficial dermis containing thin, regular elastic fibers oriented parallel to the dermal-epidermal junction. Orcein staining. Bar = 100 µm.

## Data Availability

Not applicable.

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
