# Peer review of "Auricular Non-Epithelial Tumors with Solar Elastosis in Cats: A Possible UV-Induced Pathogenesis"

_vetsci, 2022, doi:10.3390/vetsci9020034_

Round 1

Reviewer 1 Report

The paper is clear and compares histological samples from three groups of feline patients, in order to describe feline auricular non-epithelial cutaneous neoplasms associated with typical UV-induced cutaneous lesions and solar elastosis.

English language revision is advised, to correct some lapses such as “Tissues samples” in line 197.

Rewrite lines 56 to 59 (put ; to clarify studies in different species, for example)

Line 60 add HAS meaning

Line 75 “from ten”

Line 269-271 lacking reference

Author Response

Reviewer 1

The paper is clear and compares histological samples from three groups of feline patients, in order to describe feline auricular non-epithelial cutaneous neoplasms associated with typical UV-induced cutaneous lesions and solar elastosis.

English language revision is advised, to correct some lapses such as “Tissues samples” in line 197.

Response: Thanks for your comments. We corrected the mistyping (L 204).

Rewrite lines 56 to 59 (put ; to clarify studies in different species, for example)

Response: As suggested, we put ; to separate various studies in the different species (LL 58-60).

Line 60 add HAS meaning

Response: Thanks for the suggestion. We preferred to substitute directly the acronymous with the extended word in this section (L 61).

Line 75 “from ten”

Response: The space was added (L 76).

Line 269-271 lacking reference

Response:  Thanks for the comment. The reference was the same of the following sentence. However, we believe that it was necessary to add literature at this point, so we did it and changed accordingly the following sentences to avoid repetitions (LL 272 – 276).

Reviewer 2 Report

The manuscript "Auricular non-epithelial tumors with solar elastosis in cats:..." describes an interesting study on the possible role of UV exposure in the pathogenesis of non-epithelial tumor of the auricular pinna in cats.

The study is original because, generally, papers on the damage of UV exposure focus epithelial auricular cancers. Moreover, the hypothesis of a  possible UV role in the pathogenesis of non-epithelial auricular cancers in cats is consistently demonstrated in the study by  the presence of histological changes related to chronic solar irradiation in the skin adjacent to the cancers examined. 

The number of cases considered is relatively low, but non-epithelial auricular tumors are less common of the epithelial ones. Stated this, the tumors examined could be enough on my opinion.

All the paragraphs of the manuscript are well written (minor spelling errors/mistyping must be corrected - see below).

Introduction is very addressed and gives a consistent background, Mat & meth are appropriated (only a suggestion - below), results are encouraging,  consistent and supported by good images. The discussion is complete, addressed to the results, pointing out the  importance of the study but also some doubts and the necessity of further studies on these tumors that are less common than the epithelail ones but  represent an intriguing field of research

The manuscript could be suitable for publication after minor revision:

Line 75: fromten - mistyping - add a space between "from" and "ten"

Line 88: before "embedded in paraffin" insert "dehydatred through graded alcohols, claryfied in xylene" (or other reagent you used)

Line 118: use "neoplastic" instead of "tumoral"

Author Response

Reviewer 2

The manuscript "Auricular non-epithelial tumors with solar elastosis in cats:..." describes an interesting study on the possible role of UV exposure in the pathogenesis of non-epithelial tumor of the auricular pinna in cats.

The study is original because, generally, papers on the damage of UV exposure focus epithelial auricular cancers. Moreover, the hypothesis of a  possible UV role in the pathogenesis of non-epithelial auricular cancers in cats is consistently demonstrated in the study by  the presence of histological changes related to chronic solar irradiation in the skin adjacent to the cancers examined. 

The number of cases considered is relatively low, but non-epithelial auricular tumors are less common of the epithelial ones. Stated this, the tumors examined could be enough on my opinion.

All the paragraphs of the manuscript are well written (minor spelling errors/mistyping must be corrected - see below).

Introduction is very addressed and gives a consistent background, Mat & meth are appropriated (only a suggestion - below), results are encouraging,  consistent and supported by good images. The discussion is complete, addressed to the results, pointing out the  importance of the study but also some doubts and the necessity of further studies on these tumors that are less common than the epithelail ones but  represent an intriguing field of research

The manuscript could be suitable for publication after minor revision:

Line 75: fromten - mistyping - add a space between "from" and "ten"

Response: thanks for your comment. The space was added (L 76).

Line 88: before "embedded in paraffin" insert "dehydatred through graded alcohols, claryfied in xylene" (or other reagent you used)

Response: The sentence was corrected in the revised manuscript (L 89-90).

Line 118: use "neoplastic" instead of "tumoral"

Response: The adjective was changed in the revised manuscript (L 124).